# Amidase and lysozyme dual functions in TseP reveal a new family of chimeric effectors in the type VI secretion system

Zeng-Hang Wang[1,2], Ying An[2], Ting Zhao[3,4], Tong-Tong Pei[2], Dora Yuping Wang[2,5], Xiaoye Liang[2], Wenming Qin[3]*, Tao Dong[2]*

[1]State Key Laboratory of Microbial Metabolism, Joint International Research Laboratory of Metabolic & Developmental Sciences, School of Life Sciences and Biotechnology, Shanghai Jiao Tong University, Shanghai, China; [2]Department of Immunology and Microbiology, School of Life Sciences, Guangming Advanced Research Institute, Southern University of Science and Technology, Shenzhen, China; [3]National Facility for Protein Science in Shanghai, Shanghai Advanced Research Institute, Chinese Academy of Sciences, Shanghai, China; [4]School of Pharmaceutical Sciences, Wuhan University, Wuhan, China; [5]Department of Physiology, University of Toronto, Toronto, Canada

**\*For correspondence:**
qinwenming@sari.ac.cn (WQ);
dongt@sustech.edu.cn (TD)

**Competing interest:** The authors declare that no competing interests exist.

## eLife Assessment

This **important** study describes how a single effector of the Type Six Secretion System (T6SS) has two distinct functions, which may contribute to bacterial survival and the development of novel antibacterials. The authors utilized various methods in biochemistry, microbiology, and microscopy to produce **convincing** data supporting their claims about the protein's function; however, they could clarify the implications for non-experts to enhance the accessibility of this work. This manuscript is of interest to those studying T6SS, particularly those interested in effectors and bacterial enzymes.

**Abstract** Peptidoglycan (PG) serves as an essential target for antimicrobial development. An overlooked reservoir of antimicrobials lies in the form of PG-hydrolyzing enzymes naturally produced for polymicrobial competition, particularly those associated with the type VI secretion system (T6SS). Here, we report that a T6SS effector TseP, from *Aeromonas dhakensis*, represents a family of effectors with dual amidase-lysozyme activities. In vitro PG-digestion coupled with LC-MS analysis revealed the N-domain's amidase activity, which is neutralized by either catalytic mutations or the presence of the immunity protein TsiP. The N-domain, but not the C-domain, of TseP is sufficient to restore T6SS secretion in T6SS-defective mutants, underscoring its critical structural role. Using pull-down and secretion assays, we showed that these two domains interact directly with a carrier protein VgrG2 and can be secreted separately. Homologs in *Aeromonas hydrophila* and *Pseudomonas syringae* exhibited analogous dual functions. Additionally, N- and C-domains display distinctive GC contents, suggesting an evolutionary fusion event. By altering the surface charge through structural-guided design, we engineered the TseP[C4+] effector that successfully lyses otherwise resistant *Bacillus subtilis* cells, enabling the T6SS to inhibit *B. subtilis* in a contact-independent manner. This research uncovers TseP as a new family of bifunctional chimeric effectors targeting PG, offering a potential strategy to harness these proteins in the fight against antimicrobial resistance.

## Introduction

Antimicrobial resistance (AMR) is a critical global health crisis, with bacterial AMR implicated in approximately 4.95 million deaths in 2019 and predictions estimating up to 10 million deaths per year by 2050 (*Murray et al., 2022*; *O'Neill, 2016*). Developing novel antimicrobials is thus of paramount importance but also a highly challenging task. Bacterial cell walls, mainly composed of peptidoglycan (PG) and crucial for survival, serve as a primary target for antimicrobials. However, the discovery of conventional PG-inhibiting small molecules is impeded by fewer choices and increasing resistance (*Bonomo, 2017*; *Drawz and Bonomo, 2010*). The naturally occurring PG-lysing enzymes, secreted by bacteria for competition in polymicrobial environments (*Brooks et al., 2013*; *Dong et al., 2013*; *Russell et al., 2011*; *Whitney et al., 2017*), remain underexplored for AMR treatment.

As a widespread molecular weapon in Gram-negative bacteria, the type VI secretion system (T6SS) can directly inject PG-lysing enzymes and other antibacterial toxins into neighboring microbes and anti-eukaryotic toxins to yeast and mammalian cells (*Russell et al., 2011*; *Hood et al., 2010*; *Le et al., 2021*; *Ma et al., 2009*; *Mougous et al., 2006*; *Pei et al., 2022*; *Pukatzki et al., 2006*; *Trunk et al., 2018*). The conserved structure of T6SS consists of a double tubular structure, a baseplate TssEFGK complex, and a transmembrane TssJLM complex (*Basler et al., 2012*; *Brunet et al., 2015*; *Ho et al., 2014*; *Wang et al., 2019*). A spear-like inner tube, made of stacks of Hcp hexamers, is ejected outward upon contraction of an outer sheath made of stacks of TssB/C hexamers, and a VgrG-PAAR spike complex sitting atop the Hcp tube is delivered concomitantly (*Mougous et al., 2006*; *Pukatzki et al., 2006*; *Shneider et al., 2013*). Although the ejected Hcp spear can penetrate neighboring bacterial cells, T6SS-elicited damages are largely dependent on the secreted effectors rather than the penetration (*Kamal et al., 2020*; *Liang et al., 2019*; *Vettiger and Basler, 2016*). Numerous T6SS effectors have been predicted in Gram-negative bacteria, and known effectors display diverse functions including damaging the cell wall and membrane, DNase, metal scavenging, and targeting host cell actin cytoskeleton (*Dong et al., 2013*; *Liang et al., 2015*; *Pukatzki et al., 2007*; *Russell et al., 2012*; *Salomon et al., 2014*; *Si et al., 2017*; *Whitney et al., 2015*; *Yu et al., 2024*; *Das et al., 2024*). Despite the functional diversity of T6SS effectors, PG-targeting effectors seem to be most commonly associated with the T6SSs.

The conserved structure of PG comprises repeating disaccharide units of *N*-acetylglucosamine (NAG) and *N*-acetylmuramic acid (NAM), with a short peptide (most commonly ₗ-Ala-_D_-Glu-*m*-DAP-_D_-Ala-_D_-Ala) attached to each NAM (*Vollmer et al., 2008*). Adjacent peptide chains of PG are cross-linked, forming a continuous network that envelops bacterial cells (*Vollmer et al., 2008*). T6SS effectors known to target PG fall into two categories, T6SS amidase/endopeptidase (Tae) families cleaving the peptides and T6SS glycoside hydrolase (Tge) families cleaving the glycan backbone. Tae effectors have been found to cleave bonds between _D_-Glu and *m*DAP, between *m*DAP and _D_-Ala, and between NAM and ₗ-Ala (*Russell et al., 2011*; *Russell et al., 2012*; *Ma et al., 2018*; *Hernandez et al., 2020*). Tge effectors possess β-(1-4)-*N*-acetyl-muramidase activities and *N*-acetylglucosaminidase activities (*Brooks et al., 2013*; *Dong et al., 2013*; *Russell et al., 2012*; *Whitney et al., 2013*). The recent discovery of Tse4 in *Acinetobacter baumannii*, cleaving both the glycan strand and the cross-linked peptide cross-bridge, suggests that dual enzymatic activities could be found in a single effector (*Le et al., 2021*). However, such bifunctionality among effectors is rare.

As a ubiquitous waterborne pathogen, *Aeromonas dhakensis* can cause soft tissue infection and bacteremia in fish and humans (*Chen et al., 2016*; *Kitagawa et al., 2020*). The T6SS of *A. dhakensis* type strain SSU is active under laboratory conditions and can secrete three known antimicrobial effectors, a membrane-targeting TseC, a lysozyme-like TseP, and a nuclease TseI (*Liang et al., 2015*; *Liang et al., 2023*; *Liang et al., 2021b*; *Pei et al., 2020*; *Suarez et al., 2008*). Our previous findings have established that TseP serves a structural role because its expression can restore the T6SS secretion in the Δ*3eff* mutant lacking all three effector genes, and the C-terminal domain, TseP^C, possesses lysozyme activities (*Liang et al., 2021b*). However, the function of the N-terminal domain, TseP^N, which lacks any recognizable conserved motif, has remained elusive. This study unveils that TseP^N is capable of cleaving the peptide link between NAM and ₗ-Ala as an amidase. Both TseP^N and TseP^C can be secreted separately by the T6SS, but only TseP^N can compensate for T6SS defects, indicating its critical role as a structural component. Additionally, TseP^N exhibits approximately 10% higher GC content than TseP^C, and homologs of TseP^N and TseP^C exist independently in divergent species. This disparity in GC content and the presence of domain-specific homologs suggest an evolutionary

fusion event. We further show that TseP$^C$ can be engineered to hydrolyze resistant Gram-positive *Bacillus subtilis* cells, conferring the T6SS with the capability to kill Gram-positive bacteria in a contact-independent manner. This research identifies a novel class of amidase-lysozyme bifunctional effectors, provides insights into effector evolution, and highlights the potential of engineering these effectors as antimicrobials.

## Results

### TseP$^N$ and TseP$^C$ can be independently secreted by the T6SS

To delineate the regions of TseP crucial for T6SS-mediated secretion, we constructed two TseP truncations, TseP$^N$ and TseP$^C$, and then expressed them in the $\Delta 3eff$ mutant using the pBAD arabinose-inducible vectors. Consistent with our previous results (*Liang et al., 2021b*), protein secretion assays show that the $\Delta 3eff$ mutant, akin to the T6SS null $\Delta vasK$, failed to secrete Hcp (*Figure 1*, *Figure 1—figure supplement 1*). Complementation with TseP and TseP$^N$, but not TseP$^C$, restored Hcp secretion, despite at a reduced level relative to the wild-type (*Figure 1A*). We next tested whether the T6SS assembly is restored using a chromosomal sheath-labeled construct TssB-sfGFP (*Liang et al., 2021b*). Fluorescence microscopy analysis showed that complementation with both TseP and TseP$^N$, but not the TseP$^C$, restored sheath assembly (*Figure 1B and C*). Because both Hcp secretion and sheath assembly are hallmarks of T6SS assembly, these findings highlight the important structural role of the TseP N-terminus.

Interestingly, we noticed that TseP$^N$, but not TseP$^C$, was secreted (*Figure 1A*). The lack of TseP$^C$ secretion might be due to a defective T6SS. To test this, we expressed different plasmid-borne TseP constructs in a panel of SSU mutant strains. Western blot analysis not only confirmed the secretion of TseP and TseP$^N$ but also detected the secretion of TseP$^C$, albeit to a lesser extent, in both wild-type and the $\Delta tseP$ cells (*Figure 1D*). Considering that VgrG2 is the carrier protein for TseP delivery (*Liang et al., 2021b*), we next tested whether TseP$^N$ and TseP$^C$ could interact with VgrG2. Pull-down analysis showed direct interaction of VgrG2 with TseP$^N$ and TseP$^C$, in contrast to the sfGFP and MBP negative controls (*Figure 1E*). These results indicate that TseP$^N$ and TseP$^C$ can be secreted independently by the T6SS.

### TseP$^N$ is a Zn$^{2+}$-dependent amidase

When analyzing TseP-mediated PG-hydrolysis products using ultraperformance liquid chromatography coupled with mass spectrometry (UPLC-MS), we detected two distinctive peaks and determined their corresponding products to be NAM-NAG disaccharides and Tri/Tetra peptides (*Figure 2A and B*), the latter suggesting an amidase activity. The released peptides were also detected in samples treated with TseP$^N$ and the catalytically inactive C-terminal lysozyme mutant TseP$^{E663A}$ (*Liang et al., 2021b*), but not with TseP$^C$ or its inactive TseP$^{C-E663A}$ mutant, indicating that the amidase activity is attributed to TseP$^N$.

Sequence comparison with known amidases suggests that the TseP$^N$ contains a predicted Zn$^{2+}$-binding domain with several conserved residues H19, H23, H109, H339, H359, and E539 (*Figure 2C*). To test the effect of these residues on the TseP$^N$ amidase, we replaced these sites with alanine using site-directed mutagenesis and then purified a set of TseP$^N$ mutant proteins. Subsequent PG-digestion assays, coupled with UPLC-MS analysis, showed significantly reduced amidase activity in these mutants compared to TseP$^N$ (*Figure 2D*).

To test the role of metal ions for TseP's amidase activity, we added the chelating agent EDTA to the PG-digestion mixture, which abolished the amidase activity (*Figure 2E*). To further identify the specific metal ion involved, we also purified TseP in the presence of EDTA and then supplemented it with Zn$^{2+}$, Ca$^{2+}$, and a combination of both. PG-digestion assays revealed that adding Zn$^{2+}$, but not Ca$^{2+}$, restored the amidase function (*Figure 2E*).

Given the dual functionality of TseP, we sought to determine if TsiP, known as the immunity protein neutralizing the lysozyme activity, could inhibit the amidase, by mixing purified TsiP with TseP, TseP$^N$, and TseP$^C$ in varying molar ratios. Enzymatic assays using purified PG showed that TsiP could effectively inhibit both amidase activity and PG-hydrolysis activity of TseP, TseP$^N$, and TseP$^C$ (*Figure 2F*, *Figure 2—figure supplement 1*). Additionally, pull-down assays showed that TsiP interacts with

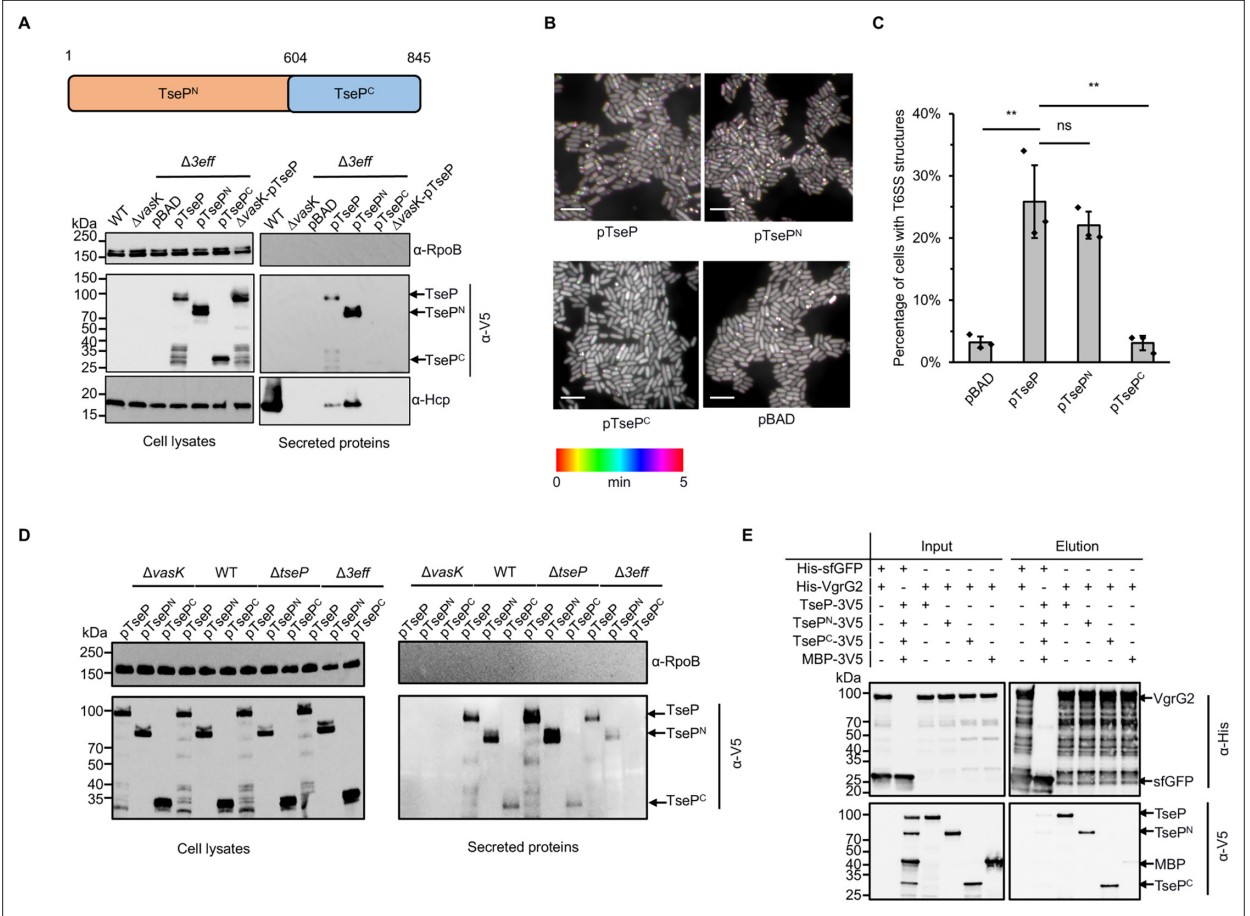

**Figure 1.** Structural roles and independent secretion of TseP domains. (**A**) Secretion analysis of TseP, TseP^N, and TseP^C in the SSU triple effector deletion mutant (Δ*3eff*). A schematic of the TseP N-terminus (TseP^N, 1–603aa) and C-terminus (TseP^C, 604–845aa) is depicted at the top. Hcp serves as a positive control for type VI secretion system (T6SS) secretion. Hcp, RpoB, and 3V5-tagged TseP proteins were detected using specific antibodies. (**B**) Time-lapse imaging of VipA-sfGFP signals in the Δ*3eff* mutant complemented with different TseP variants. Each sample was captured every 10 s for 5 min and temporally color-coded. Color scale used to temporally color-code the VipA-sfGFP signals is shown at the bottom. A 30×30 μm² representative field of cells is shown. Scale bars, 5 μm. (**C**) Statistical analysis of T6SS sheath assemblies in the Δ*3eff* mutant complemented with different TseP variants. Error bars indicate the mean ± standard deviation of three biological replicates, and statistical significance was calculated using a two-tailed Student's *t*-test. ns, not significant; **, p<0.01. (**D**) Secretion analysis of TseP, TseP^N, and TseP^C in SSU wild-type, Δ*vasK*, Δ*tseP*, and Δ*3eff* mutants. For (**B**) and (**D**) TseP, TseP^N, and TseP^C were tagged with a 3V5 C-terminal tag and expressed on pBAD vectors. RpoB serves as an equal loading and autolysis control. (**E**) Pull-down analysis of VgrG2 with TseP, TseP^N, and TseP^C. His-tagged VgrG2 and 3V5-tagged TseP, TseP^N, or TseP^C were used. His-tagged sfGFP and 3V5-tagged MBP were used as controls.

The online version of this article includes the following source data and figure supplement(s) for figure 1:

**Source data 1.** Original files for Western blot analysis displayed in *Figure 1A, D, and E*.

**Source data 2.** PDF file containing original Western blots for *Figure 1A, D, and E*.

**Figure supplement 1.** Staining SDS-PAGE of secretion of TseP, TseP^N, and TseP^C in the SSU triple effector deletion mutant (Δ*3eff*).

**Figure supplement 1—source data 1.** Original files for SDS-PAGE analysis displayed in *Figure 1—figure supplement 1*.

**Figure supplement 1—source data 2.** PDF file containing original SDS-PAGE for *Figure 1—figure supplement 1*.

not only TseP^C but also TseP^N (*Figure 2G*). Collectively, these results indicate that TseP^N is a Zn^{2+}-dependent amidase.

## Amidase activity is not required for T6SS assembly or lysozyme function

To determine whether the amidase activity is important for the structural role of TseP^N, we expressed these amidase-inactive mutants in the Δ*3eff* TssB-sfGFP mutant and tested T6SS-related functions.

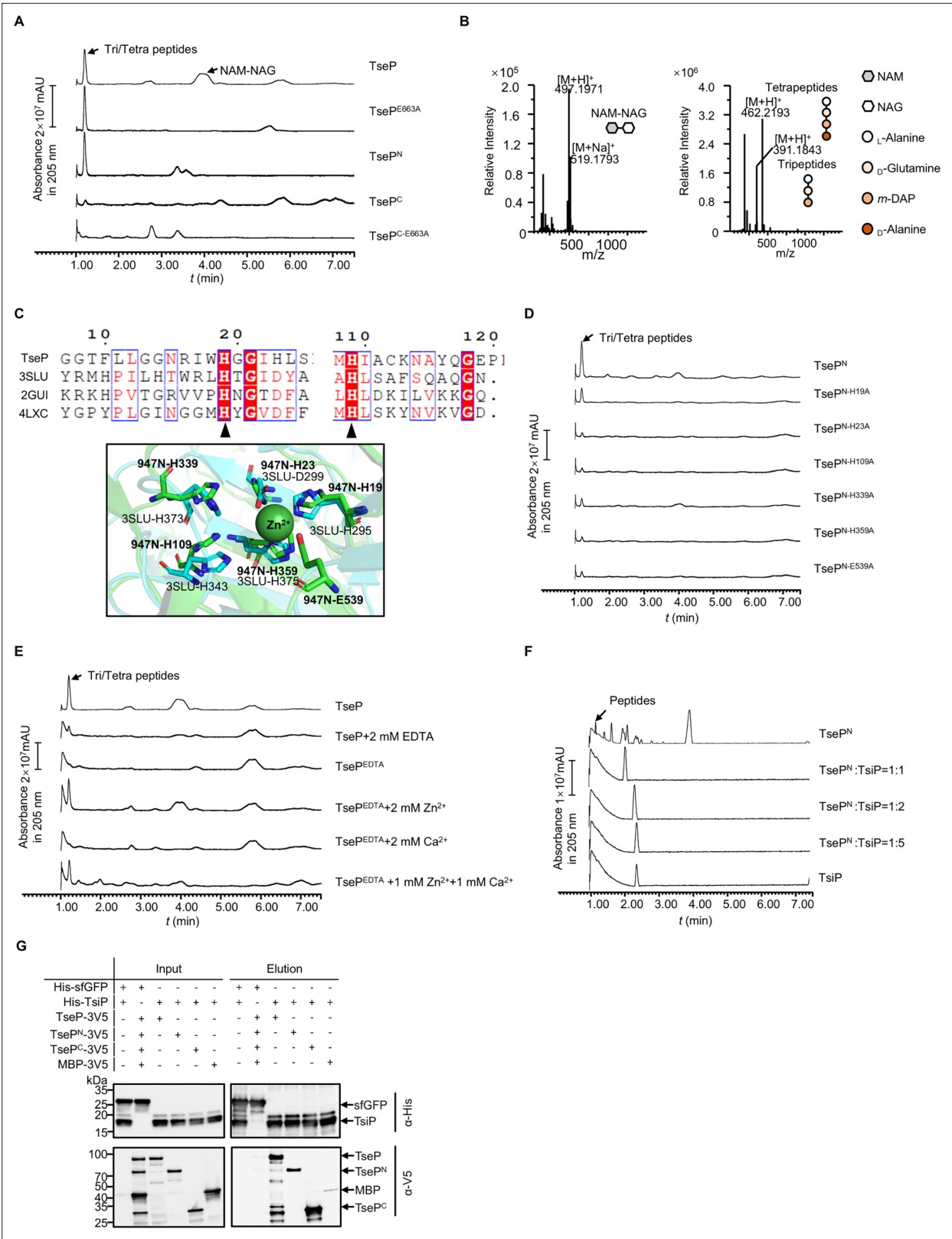

**Figure 2.** Functional analysis of TseP reveals an amidase activity. (**A**) In vitro amidase activity of the TseP, TseP[N], TseP[C], and the lysozyme inactivated mutants TseP[E663A] and TseP[C-E663A]. Cell-wall digestion products after incubation with the TseP or its mutants were analyzed by the ultraperformance liquid chromatography coupled with mass spectrometry (UPLC/MS). (**B**) MS analysis of cell-wall digestion products (*N*-acetylmuramic acid [NAM]-NAC, tetrapeptides, and tripeptides) following treatment with TseP. (**C**) Protein sequence alignment of the TseP amidase domain with other amidase

*Figure 2 continued on next page*

*Figure 2 continued*

homologs (top), and the structural superimposition of the TseP amidase domain and 3SLU (bottom). Cartoon representations of TseP and 3SLU are shown in green and cyan, respectively. The key residues involved in $Zn^{2+}$ binding are shown in a stick model, and the zinc ion is indicated by the green sphere. (**D**) In vitro amidase activity of $TseP^N$ and its amidase site mutated variants. (**E**) In vitro amidase activity of TseP under cationic conditions of 2 mM EDTA, $Zn^{2+}$, $Ca^{2+}$, or a combination of both cations. TseP represents protein purified without EDTA treatment while $TseP^{EDTA}$ refers to protein purified in the presence of EDTA. (**F**) Peptidoglycan (PG) digestion analysis of $TseP^N$ with or without TsiP. The immunity protein TsiP was incubated with $TseP^N$ on ice for 12 hr before being mixed with PG. Products in (**D**), (**E**), and (**F**) were analyzed by UPLC-quadrupole time of flight (QTOF) mass spectrometry. (**G**) Pull-down analysis of TsiP with TseP, $TseP^N$, and $TseP^C$. His-tagged TsiP and 3V5-tagged TseP, $TseP^N$, or $TseP^C$ were used. His-tagged sfGFP and 3V5-tagged MBP were used as controls.

The online version of this article includes the following source data and figure supplement(s) for figure 2:

**Source data 1.** Original files for Western blot analysis displayed in *Figure 2G*.

**Source data 2.** PDF file containing original Western blots for *Figure 2G*.

**Figure supplement 1.** TsiP inhibits both the amidase and lysozyme activities of TseP.

Secretion analysis revealed that expression of $TseP^{H339A}$, but not the other mutants, restored Hcp secretion similar to wild-type TseP (*Figure 3A*). As control, we examined the expression of these TseP variants and found comparable signals in the cell lysates and secretion samples. Observations of TssB-sfGFP sheath assembly in the Δ*3eff* TssB-sfGFP mutant further supported these findings, with the $TseP^{H339A}$ strain showing dynamic assembly of T6SS similar to the strain expressing the wild-type protein (*Figure 3B*, *Figure 3—figure supplement 1A and C*). Further analysis through bacterial

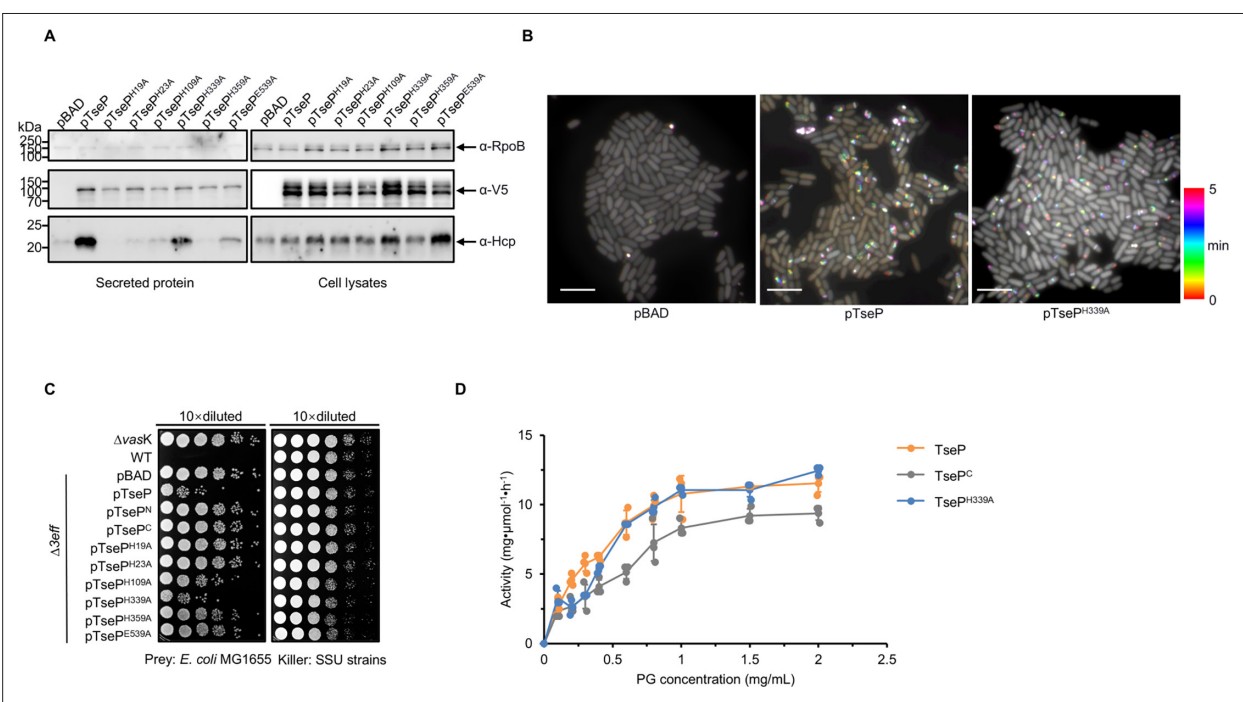

**Figure 3.** Amidase activity of TseP is not essential for type VI secretion system (T6SS) assembly or lysozyme function. (**A**) Secretion analysis of Hcp in the Δ*3eff* mutant complemented with different TseP variants. RpoB serves as an equal loading and autolysis control. Hcp, RpoB, and 3V5-tagged TseP proteins were detected using specific antibodies. (**B**) Time-lapse imaging of VipA-sfGFP signals in the Δ*3eff* mutant complemented with TseP or its amidase-inactive mutant $TseP^{H339A}$. Each sample was captured every 10 s for 5 min and temporally color-coded. Color scale used to temporally color-code the VipA-sfGFP signals is shown at the right. A 30×30 $\mu m^2$ representative field of cells is shown. Scale bars, 5 μm. (**C**) Competition analysis of the Δ*3eff* mutant complemented with different TseP variants. Competition assays were repeated once. (**D**) Glycoside hydrolase activity of the TseP, $TseP^C$, and amidase-inactive mutant $TseP^{H339A}$. The error bars indicate the mean ± standard deviation of three biological replicates.

The online version of this article includes the following source data and figure supplement(s) for figure 3:

**Source data 1.** Original files for Western blot analysis displayed in *Figure 3A*.

**Source data 2.** PDF file containing original Western blots for *Figure 3A*.

**Figure supplement 1.** The amidase activity of TseP is not required for either type VI secretion system (T6SS) assembly or lysozyme function.

competition assays, with *Escherichia coli* MG1655 strain as prey, mirrored these results (*Figure 3C*). The expression of TseP[H339A] and TseP showed comparable killing abilities while the expression of other mutants demonstrated severely impaired killing activities as the T6SS-defective Δ*3eff* (*Figure 3C*). Although it is unclear why the other mutations failed to complement, the H339A amidase-inactive mutation exhibiting a similar complementary role to wild-type TseP suggests that the amidase activity is not required for T6SS assembly.

Additionally, we tested whether there is any synergistic effect between amidase and the TseP[C]-mediated lysozyme activities by comparing PG degradation treated with TseP, TseP[C], and TseP[H339A]. Results showed that TseP exhibited higher glycoside hydrolase activities than the TseP[C] alone, but there was no difference between TseP and the TseP[H339A] mutant (*Figure 3D*, *Figure 3—figure supplement 1B*). This indicates that amidase and lysozyme activities can each operate as functional modules within the protein.

## TseP homologs exhibit similar dual functions

Homolog searching analysis reveals a broad distribution of TseP homologs across Gram-negative bacteria (*Figure 4A*, *Figure 4—figure supplement 1*). To examine the dual functionality of these homologs, we selected two representative proteins that have not been previously characterized: AHA_1849 from *Aeromonas hydrophila* which is a closely related homolog from the same genus, and PSPTO_5204 from *Pseudomonas syringae*, a more distantly related homolog. To determine whether AHA_1849 and PSPTO_5204 share amidase and lysozyme activities as TseP, we expressed and purified their respective N- and C-terminal truncated mutants, AHA_1849[N] (1–640aa), AHA_1849[C] (641–851aa), PSPTO_5204[N] (1–343aa), and PSPTO_5204[C] (344–588aa) (*Figure 4—figure supplement 1*). We also constructed inactivating mutations in their predicted catalytic residues for each protein to serve as controls (*Figure 4—figure supplement 1*). Subsequent PG-hydrolysis and UPLC-MS analysis confirmed that both AHA_1849 and PSPTO_5204 exhibit dual amidase and lysozyme activities, capable of cleaving both peptide and glycan chains (*Figure 4B and C*).

Additionally, by analyzing the GC-contents of the two *Aeromonas* homologs, we found that the GC-content of TseP[N] is about 10% higher than that of TseP[C] and 10% lower than the upstream VgrG (*Figure 4D*). This suggests that these two domains might have been horizontally transferred and subsequently fused during evolution. To explore this hypothesis further, we searched for homolog proteins with similar structures to either TseP[N] or TseP[C] in the Foldseek Search AFDB50 database (*van Kempen et al., 2024*), yielding 1000 proteins with comparable predicted structures (*Supplementary file 2* and *Supplementary file 3*). We identified TseP[N]-only homologs in various *Aeromonas* species, including a specific homolog in *A. schubertii* possessing only the TseP[N] domain, whereas TseP[C] homologs were discovered in several other species, such as *Fulvimonas soli* and *Vibrio campbellii* (*Figure 4E*). The comparison of their predicted structures reveals high similarity (*Figure 4F*, *Supplementary file 1b and c*). Considering the GC-content difference, we postulate that TseP[N] and TseP[C] domains may have been fused through a recombination event, resulting in a chimeric, dual-functional TseP.

## Crystallographic analysis of TseP[C] reveals a wide active groove

While testing TseP-mediated hydrolysis of purified PG from Gram-negative bacteria, we noticed that both TseP and TseP[C] displayed more efficient digestion than the commonly used hen-egg lysozyme (*Figure 5A*). This enhanced activity suggests that TseP may utilize a unique mechanism for cell-wall degradation. To further investigate this, we determined the X-ray crystal structure of TseP[C] to a resolution of 2.27 Å ($R_{work}$ = 0.189 and $R_{free}$ = 0.227) by molecular replacement with one molecule in the asymmetric unit (*Table 1*).

The crystal structure revealed that TseP[C] comprises a smaller lobe of seven helices and a larger lobe of ten helices, separated by a concave groove (*Figure 5B*, *Figure 5—figure supplement 1A*). The surface electrostatic potential of TseP[C], calculated using the Adaptive Poisson-Boltzmann Solver (APBS), consists of both electronegative and electropositive regions, unlike the predominantly electronegative groove of lysozyme (*Figure 5C*, *Figure 5—figure supplement 1B*). The electropositive surface of TseP[C] consists of residues R723 and K796, with an adjacent residue Y720 corresponding to W62 in lysozyme, altering which is known to affect substrate binding (*Maenaka et al., 1995*). Additionally, the previously predicted catalytic dyad (E655 and E663) was located near the groove. Although this structural arrangement resembles that of other lysozyme family members, TseP[C]

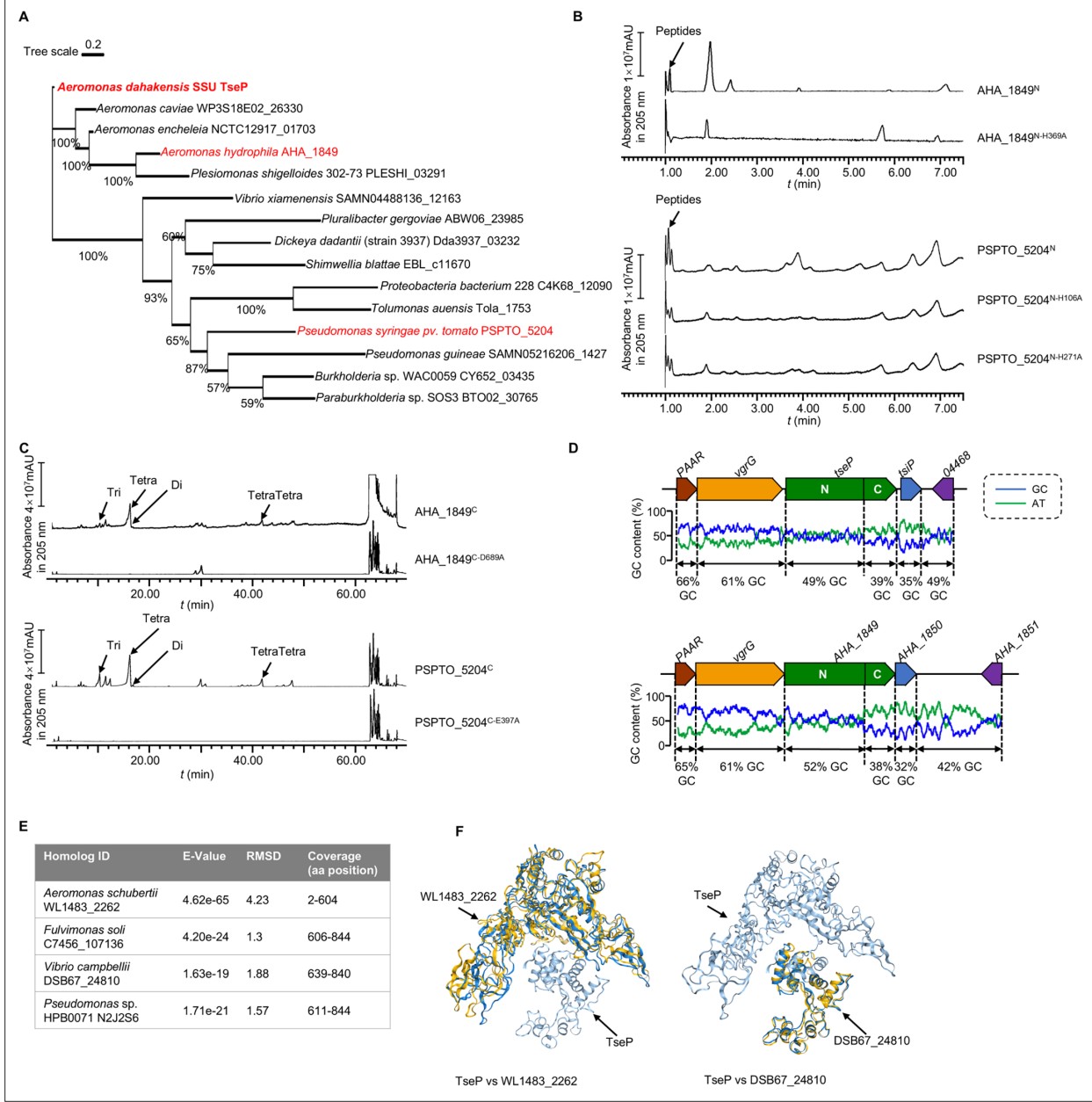

**Figure 4.** TseP homologs showed the same in vitro peptidoglycan (PG)-hydrolysis activity with the TseP. (**A**) Maximum-likelihood phylogeny of TseP homologs. Phylogeny was constructed using the IQ-tree web server with bootstrap 1000 times. Proteins tested in this study are highlighted in red. (**B**) Amidase activity analysis of TseP homologs. (**C**) In vitro PG-hydrolysis activity of the TseP homologs. Products in (**B**) and (**C**) were analyzed through ultraperformance liquid chromatography-quadrupole time of flight (UPLC-QTOF) mass spectrometry. (**D**) GC contents of the *tseP* gene cluster and *AHA_1849* gene cluster. (**E**) Summary of TseP^N and TseP^C homologs output by Foldseek Search server. (**F**) Structure alignments of TseP and homologs WL1483_2262 and DSB67_24810.

The online version of this article includes the following figure supplement(s) for figure 4:

**Figure supplement 1.** Protein sequence analysis of the TseP homologs.

exhibits a less compact conformation with a notably wider groove compared to the typical lysozyme (*Figure 5D*). Previous studies have shown that the lysozyme possesses catalytic sites at E53 and D70 in the substrate-binding groove (*Maenaka et al., 1995*), while the active residues in the TseP^C structure are E654 and E663, respectively (*Figure 5D*). To test the effect of E663 for TseP^C activity, we purified a mutant E663D (TseP^C-E663D) protein and perform PG-digestion assays. The results show that TseP^C-E663D lost the activity, suggesting the greater distance within the groove requires this glutamate residue

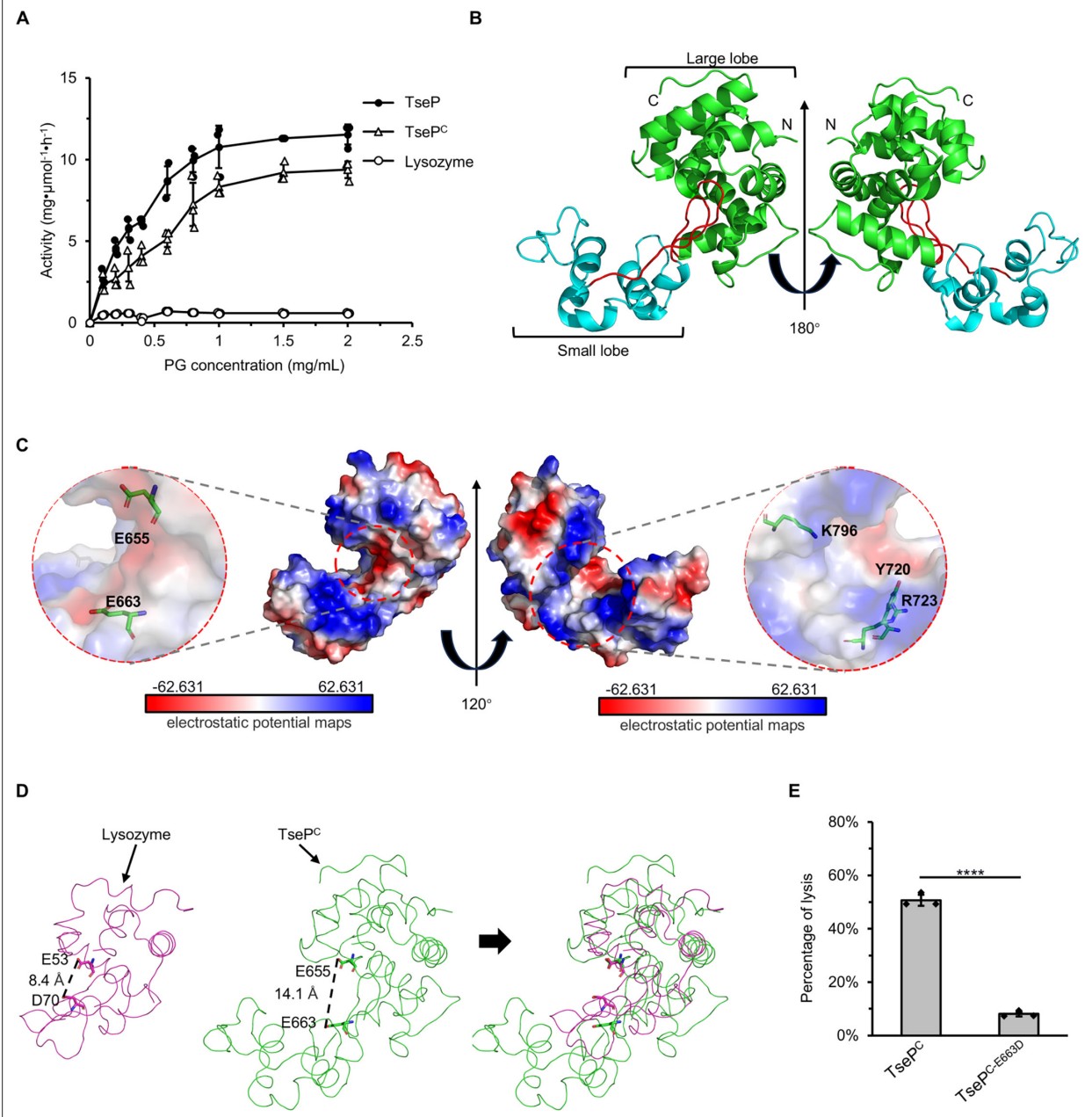

**Figure 5.** Crystal structure of TseP^C. (**A**) Enzymatic activity of TseP, TseP^C, and lysozyme in hydrolyzing purified *E. coli* peptidoglycan (PG). The error bars indicate the mean standard deviation of three biological replicates. (**B**) The overall structure of the TseP^C. The small lobe and large lobe are shown in cyan and green, respectively, with the connecting loop depicted in red. (**C**) Electrostatic potential maps of TseP^C with the Y720, R723, and K796 shown as a stick model. The electrostatic surface potentials are colored red for negative charges, blue for positive charges, and white for neutral residues. (**D**) Structural comparison of TseP^C and lysozyme (PDB ID: 1LZC). The catalytic sites are shown as a stick model. (**E**) *E. coli* PG-digestion analysis of TseP^C and TseP^C-E663D mutant. Error bars indicate the mean ± standard deviation of three biological replicates, and statistical significance was calculated using a two-tailed Student's *t*-test. ****, p<0.0001.

The online version of this article includes the following figure supplement(s) for figure 5:

**Figure supplement 1.** Structural and mutational analyses of the C-terminal domain.

**Table 1.** Data collection and refinement statistics of TseP$^C$ crystallization.

| | TseP$^C$- 8XCL |
|---|---|
| **Data collection** | |
| Space group | C 2 2 21 |
| a, b, c (Å) | 44.48 138.80 95.83 |
| α, β, γ (°) | 90.00 90.00 90.00 |
| Wavelength (Å) | 0.97852 |
| Resolution (Å) | 47.91–2.27 (2.40–2.27) |
| CC$_{1/2}$ | 0.984 (0.840) |
| Unique reflections | 25776 (4304) |
| R$_{meas}$ (%)* | 23.6 (83.6) |
| Mean I/σ (I)* | 6.1 (2.6) |
| Completeness (%)* | 95.9 (99.4) |
| Multiplicity* | 12.9 (12.0) |
| **Refinement** | |
| Resolution (Å) | 47.91–2.27 |
| R$_{work}$/R$_{free}$† | 0.189/0.227 |
| No. atoms | |
| Protein | 1910 |
| Water | 123 |
| Average B factors (Å$^2$) | |
| Protein | 31.72 |
| Water | 37.40 |
| R.m.s. deviations | |
| Bond lengths (Å) | 0.007 |
| Bond angles (°) | 0.871 |
| Ramachandran plot | |
| Favored (%) | 97.50 |
| Allowed (%) | 2.50 |
| Outliers (%) | 0.00 |

$$R_{means} = \sum_{hkl} \sqrt{n/(n-1)} \sum_{i=1}^{n} | I_i(hkl) - \langle I(hkl) \rangle | / \sum_{hkl} \sum_{i} I_i(hkl),$$ where $\langle I(hkl) \rangle$ is the mean intensity of a set of equivalent reflections.

$$R_{work} = \sum_{hkl} ||F_{obs}| - |F_{calc}|| / \sum_{hkl} |F_{obs}|,$$ where $F_{obs}$ and $F_{calc}$ are observed and calculated structure factors, respectively.

*The values in parentheses are for the outermost shell.

†R$_{free}$ is the R$_{work}$ based on 5% of the data excluded from the refinement.

(*Figure 5E*). These results collectively highlight the key structural features that may account for the increased activity of TseP$^C$ relative to the hen-egg lysozyme.

## Engineered TseP$^C$ can lyse Gram-positive cells

To test whether TseP can degrade various cell walls, we purified PG from *B. subtilis*, a model Gram-positive bacterium, for use as substrate. However, we failed to detect any PG-hydrolysis treated with TseP or TseP$^C$, in contrast to the lysozyme (*Figure 6A*). Further purification of PG with NaOH

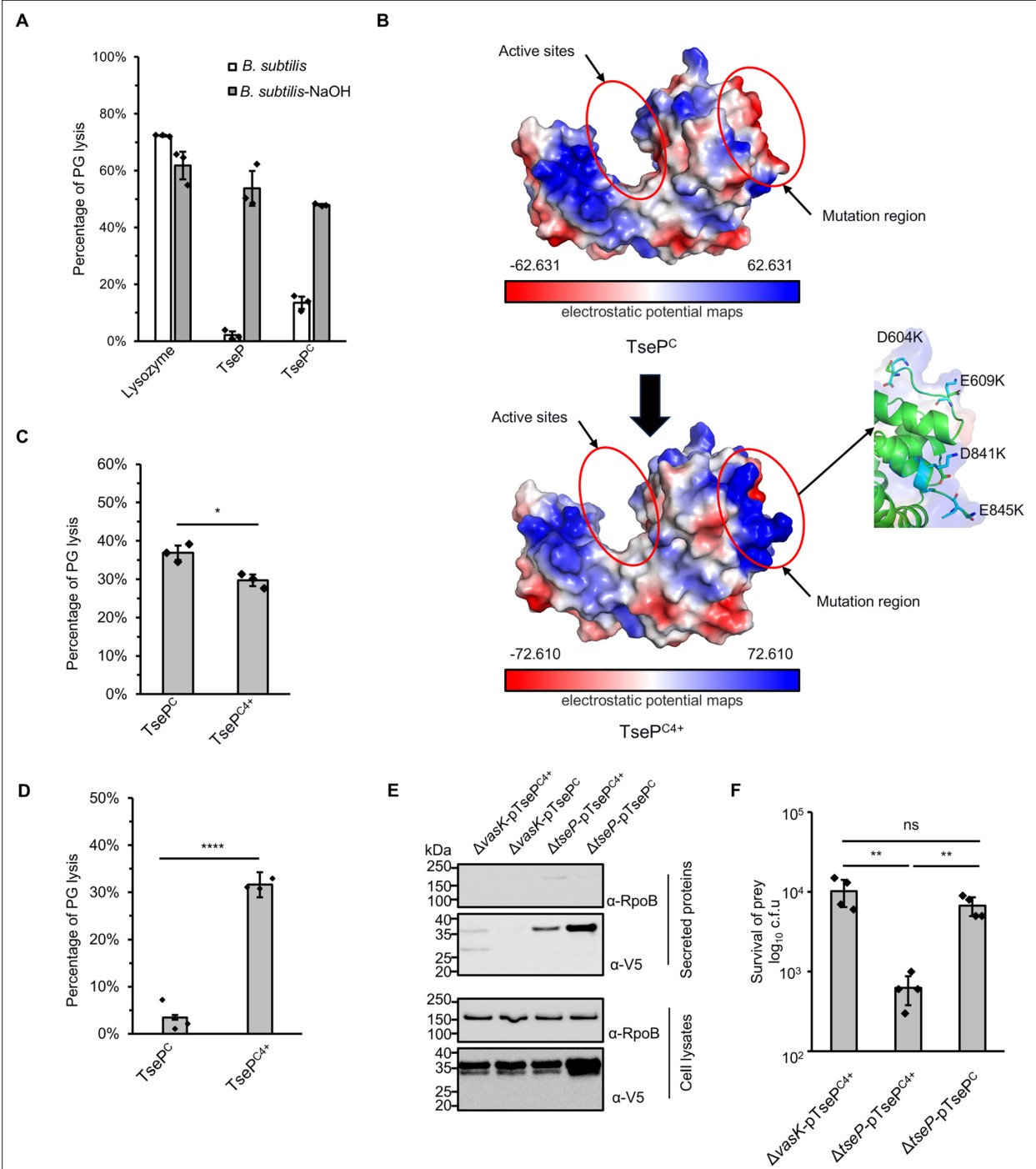

**Figure 6.** Modification of surface charge enables TseP^C to kill Gram-positive bacteria. (**A**) *B. subtilis* peptidoglycan (PG) digestion analysis of TseP and TseP^C. PG was treated with 0.25 M NaOH for 12 hr at 37°C to remove cross-linked peptides and teichoic acid. The lysis percentage was calculated by detecting the changes of $OD_{600}$ during 1 hr. The hen-egg lysozyme was used as a positive control. (**B**) Electrostatic potential maps of the TseP^C and TseP^{C4+}. The active sites and the mutation region are highlighted in red circles. The negatively charged residues D604, E609, D841, and E845 in the mutation region are shown as a stick model and colored in green, and lysines are colored in cyan. (**C**) In vitro PG-hydrolysis activity of the TseP^C and TseP^{C4+}. RBB-labeled *E. coli* PG was used as the substrate and the lysis percentage was detected by dye release. (**D**) *B. subtilis* PG-digestion analysis of TseP^C and TseP^{C4+}. Exponential phase *B. subtilis* cells ($OD_{600}$~1.0) were used as substrate. The lysis percentage was calculated by detecting the changes of $OD_{600}$ during 1 hr with the enzyme concentration at 100 nM. (**E**) Secretion analysis of TseP^C and TseP^{C4+} in the Δ*tseP* mutant. RpoB serves as an equal loading and autolysis control. RpoB and 3V5-tagged TseP^C proteins were detected using specific antibodies. (**F**) Statistical analysis of *B. subtilis* cells in

*Figure 6 continued on next page*

*Figure 6 continued*

the competition assays. Error bars of statistical analysis in (**A**), (**C**), (**D**), and (**F**) indicate the mean ± standard deviation of three biological replicates, and statistical significance was calculated using a two-tailed Student's *t*-test. ns, not significant; *, p<0.05; **, p<0.01; ****, p<0.0001.

The online version of this article includes the following source data and figure supplement(s) for figure 6:

**Source data 1.** Original files for Western blot analysis displayed in *Figure 6E*.

**Source data 2.** PDF file containing original Western blots for *Figure 6E*.

**Figure supplement 1.** EF-hand domain has no effect on TseP activity.

**Figure supplement 2.** Expression and secretion of TseP variants.

**Figure supplement 2—source data 1.** Original files for Western blot analysis displayed in *Figure 6—figure supplement 2*.

**Figure supplement 2—source data 2.** PDF file containing original Western blots for *Figure 6—figure supplement 2*.

**Figure supplement 3.** Purification of TseP variants and homologous proteins.

**Figure supplement 3—source data 1.** Original files for SDS-PAGE analysis displayed in *Figure 6—figure supplement 3*.

**Figure supplement 3—source data 2.** PDF file containing original SDS-PAGE for *Figure 6—figure supplement 3*.

treatment, removing PG-associated peptides, led to effective hydrolysis of PG by TseP and TseP$^C$, suggesting a difference in substrate accessibility between the lysozyme and TseP. By comparing protein surface electrostatic potentials between the lysozyme and TseP$^C$, we noticed a more electronegative surface patch in TseP$^C$, suggesting a potential electrostatic repulsion from negatively charged PG (*Figure 6B*, *Figure 5—figure supplement 1B*). To increase electropositive regions while minimizing impact on activity, we mutated four negative charge residues distant from the active site and obtained a quadruple mutant TseP$^{C4+}$ (*Figure 6B*). Activity assays confirmed that TseP$^{C4+}$ retained *E. coli* PG-hydrolyzing activities (*Figure 6C*). To test whether the engineered TseP$^{C4+}$ was able to lyse *B. subtilis*, we treated *B. subtilis* cells with purified TseP$^C$ and TseP$^{C4+}$ and found that cells were lysed effectively by TseP$^{C4+}$ but not TseP$^C$ (*Figure 6D*, *Figure 5—figure supplement 1C*). These results indicate that modulating the surface charge is an effective strategy to expand the target range of PG-lysing effectors.

Next, we tested whether the engineered TseP$^{C4+}$ could be delivered by the T6SS for inhibiting *B. subtilis*. Protein secretion analysis showed a significant TseP$^{C4+}$ signal in secretion samples, despite at a lower level than the wild-type TseP$^C$ (*Figure 6E*). When *B. subtilis* was co-incubated with the Δ*tseP* mutant expressing the engineered TseP$^{C4+}$ in liquid culture, survival of *B. subtilis* was significantly reduced in a T6SS-dependent manner (*Figure 6F*, *Figure 5—figure supplement 1D*). These data collectively suggest that the T6SS can be armed with engineered TseP$^{C4+}$ to gain Gram-positive killing capabilities in a contact-independent manner.

## Discussion

The T6SS plays a pivotal role in bacterial competition, using a variety of antibacterial effectors to kill competing cells. The exploration of effector functions holds the promise of unlocking a largely untapped source of novel antimicrobials, presenting innovative approaches to address the escalating AMR crisis. Here, we report that TseP represents a novel family of dual-functional chimeric effectors, exhibiting both amidase and lysozyme activities (*Figure 7*). Our study reveals that the N-terminus of TseP not only serves a critical structural role in T6SS assembly but also acts as a Zn$^{2+}$-dependent amidase. The N- and C-domains are both secreted and functionally independent, and may be subject to an evolutionary fusion event to form the chimeric full-length TseP. Such multifunctionality underscores a complex evolutionary strategy aimed at promoting T6SS-mediated competitiveness.

Among the known T6SS effectors, those targeting the bacterial PG layer represent a prevalent category. Certain T6SSs are capable of translocating multiple PG-targeting effectors. For instance, the H1-T6SS in *Pseudomonas aeruginosa* deploys Tse1 and Tse3, while *Vibrio cholerae* secretes VgrG3 and TseH (*Brooks et al., 2013*; *Dong et al., 2013*; *Russell et al., 2011*; *Hood et al., 2010*; *Altindis et al., 2015*; *Hersch et al., 2020*). Tse1 and TseH exhibit endopeptidase activities, whereas Tse3 and VgrG3 have potent lysozyme activities (*Brooks et al., 2013*; *Dong et al., 2013*; *Russell et al., 2011*; *Hood et al., 2010*; *Altindis et al., 2015*; *Hersch et al., 2020*). Yet, it remains relatively uncommon to identify effectors that possess dual functionality, capable of cleaving both the glycan chains and the

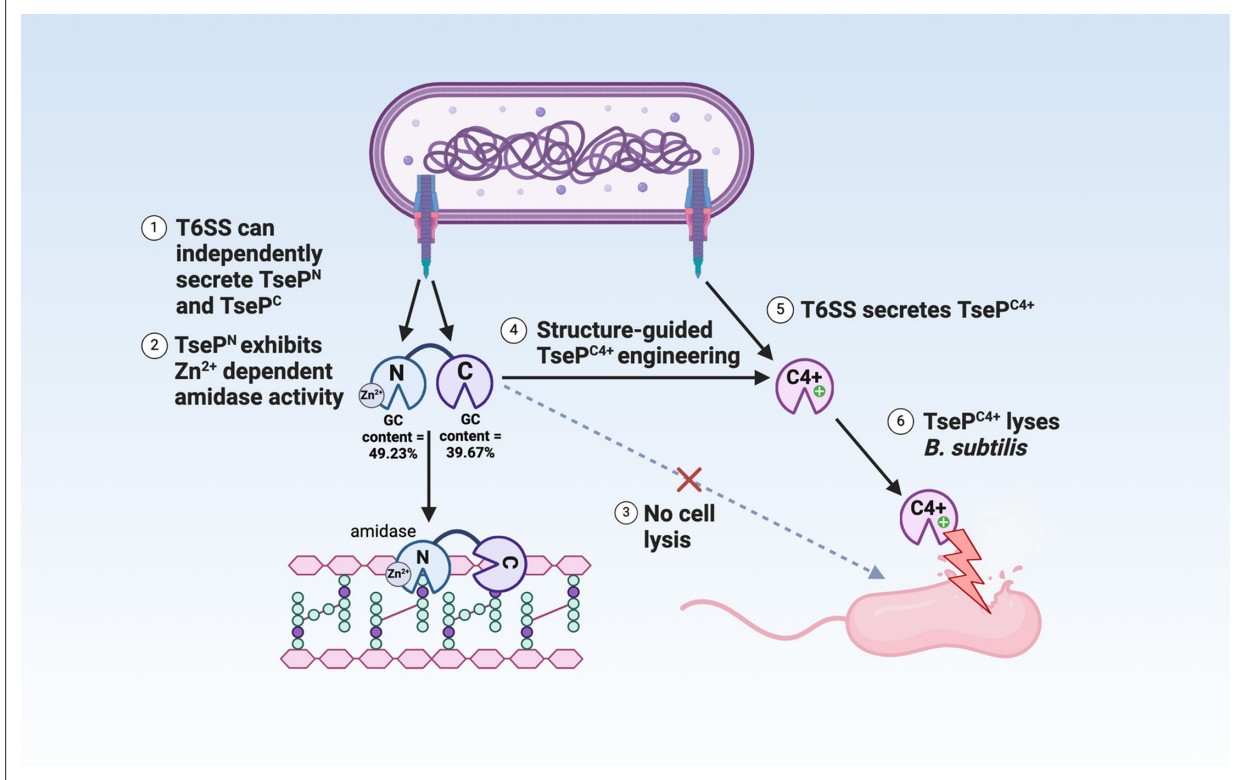

**Figure 7.** Model of TseP dual functions and engineering. This schematic illustrates the dual amidase-lysozyme activities of TseP and demonstrates an effective engineering strategy to enhance type VI secretion system (T6SS) and its effector capabilities. The T6SS can independently secrete both the N- and C-terminal domains, with multiple evidence suggesting an evolutionary fusion event. These domains interact directly with the upstream-encoded carrier protein VgrG2 for secretion. TseP[N], the N-terminal domain, functions as a $Zn^{2+}$-dependent amidase, while the C-terminal domain, TseP[C], exhibits lysozyme activities. However, in its native form, TseP[C] does not lyse Gram-positive *B. subtilis* cells. Through structural-guided design, TseP[C] was engineered to create TseP[C4+] by altering its surface charge. This modification enables TseP[C4+] to lyse *B. subtilis* cells without affecting its T6SS-dependent secretion. Consequently, the T6SS-equipped cell acquires the ability to lyse *B. subtilis* in a contact-independent manner. Given the diversity of TseP-like effectors and T6SS species, this approach holds significant potential for modulating interspecies competition and combating antimicrobial resistance.

cross-linked peptides within the bacterial cell wall. To date, the sole established example is Tse4 from *A. baumannii*, which acts both as a transglycosylase to cleave the glycan strand and as an endopeptidase to cut the peptide cross-bridge (*Le et al., 2021*; *Chou et al., 2012*; *Radkov et al., 2022*). Our enzymatic assays reveal that TseP distinguishes itself from Tse4 with dual roles as a lysozyme and an amidase, as well as a structural role for T6SS assembly. Further homology searches and biochemical validations of homolog activities substantiate that TseP constitutes a new family of dual-functional effectors.

Our data demonstrate that the N-terminus of TseP can modulate T6SS assembly, which provides further evidence that effectors play a crucial structural role in T6SS assembly (*Liang et al., 2019*; *Liang et al., 2023*; *Liang et al., 2021a*; *Wu et al., 2020*). Although this notion has now been shown in several species, the molecular details remain elusive. The assembly of a non-contractile mutant T6SS in *V. cholerae* in the absence of effectors suggests that effectors might be needed to stabilize a polymerizing T6SS (*Stietz et al., 2020*). In comparison to TseP[C], the N-terminus not only contributes to T6SS assembly but also exhibits 10% higher GC content. Additionally, both N and C-terminus can be independently secreted by the T6SS. Using structure-assisted homolog search, we have identified TseP[N] and TseP[C]-only proteins in different species. These data suggest that these two domains may result from independent effector proteins and have undergone an evolutionary fusion event.

Previous studies have shown that expressing TseP in the periplasm is highly toxic, but this toxicity is mitigated when catalytic mutations are introduced into the C-terminal lysozyme domain (*Liang et al., 2021b*). This indicates that the N-terminal amidase is either less toxic or is tightly regulated by

the concentration of $Zn^{2+}$ in the growth medium. Although the total intracellular $Zn^{2+}$ concentration during growth in Lysogeny Broth (LB) has been reported to be around 0.1 mM—approximately 100 times higher than the $Zn^{2+}$ level in the LB medium—the bioavailable (free) $Zn^{2+}$ concentration within *E. coli* cells remains exceedingly low (*Outten and O'Halloran, 2001*). A similar dependency on metal ions was observed with another cell-wall-targeting endopeptidase effector, TseH, whose activity is dependent on the environmental concentrations of $Mg^{2+}$ and $Ca^{2+}$ (*Tang et al., 2022*). Such metal ion dependencies might lead to misbelief about these effectors being cryptic or inactive, complicating the discovery and functional analysis of these important proteins.

TseP is predicted to contain an EF-hand calcium-binding domain with a 'helix-loop-helix' motif, a feature previously unreported in T6SS effectors (*Figure 6—figure supplement 1*). However, mutations introduced to this domain, aimed at exploring its role in T6SS assembly and TseP toxicity, did not reveal any critical effect on TseP functions under the conditions tested (*Figure 6—figure supplement 1*). It is possible that this EF-hand motif is active under more $Ca^{2+}$ stringent conditions or during infections. Further study is required to decipher its physiological function. Additionally, while constructing TseP variants with different residue changes, we noticed that even single residue changes could reduce the expression, secretion, or the structural role of TseP, highlighting the technical challenge of heterologous effector engineering (*Figure 3A*, *Figure 6—figure supplement 2*).

Because the outer membrane of Gram-negative bacteria generally prohibits the entry of cell-wall-lysing proteins, the relatively exposed Gram-positive bacteria are likely more amenable to such treatments. However, the T6SS, exclusively found in Gram-negative bacteria, primarily targets Gram-negative species. To date, only two T6SSs, identified in *Acidovorax citrulli* and *A. baumannii* (*Le et al., 2021*; *Pei et al., 2022*), have shown the capability to kill Gram-positive bacteria. Additionally, no Gram-positive cell wall lysing effectors have been reported; for instance, Tse4 from *A. baumannii* cannot directly lyse *B. subtilis* cultures but requires T6SS delivery. Our biochemical characterization and engineering of TseP[C] not only establishes its exceptional enzymatic activity but also demonstrates an effective strategy to acquire Gram-positive cell-lysing activities, as well as to equip the T6SS of *A. dhakensis* with a contact-independent killing ability. These results highlight the potential of antimicrobial effector engineering in addressing AMR.

In conclusion, our study unveils a new family of dual-functional chimeric T6SS effectors and demonstrates the potential for bioengineering these proteins to broaden their antimicrobial spectrum. The engineered TseP[C] represents the first effector variant capable of directly lysing *B. subtilis* cultures, paving the way for engineering the diverse set of T6SS cell-wall-targeting effectors. The urgent need for new antimicrobials necessitates the exploration of innovative strategies. The insights from our research into T6SS and its diverse effectors signal promising directions for future work in microbial biology and antimicrobial therapy. When enhanced through protein engineering, these enzymes have the potential to address bacterial infections not only in medical settings but also in food safety and animal health.

# Materials and methods

## Bacterial strains and growth conditions

All strains and plasmids used in this study are displayed in *Supplementary file 1a*. Strains were routinely cultivated in LB ([wt/vol] 1% tryptone, 0.5% yeast extract, 0.5% NaCl) under aerobic conditions at 37°C. Antibiotics were supplemented at the following concentrations: streptomycin (100 μg/ml), kanamycin (50 μg/ml), gentamicin (20 μg/ml), chloramphenicol (25 μg/ml for *E. coli*, 2.5 μg/ml for SSU). All plasmids were constructed using standard molecular techniques, and their sequences were verified through Sanger sequencing. All strains and plasmids are available upon request.

## Protein secretion assay

Secretion assays were performed as previously described (*Liang et al., 2021b*). Briefly, cells were grown in LB to $OD_{600}$=1 and collected by centrifugation at 2500×*g*. The pellets were resuspended in fresh LB and incubated at 28°C for 2 hr with 0.01% (wt/vol) L-arabinose. After incubation, cells were pelleted again by centrifugation at 2500×*g* for 8 min. Cell pellets were then resuspended in SDS-loading dye and used as cell lysate samples. The supernatants were centrifuged again at 12,000×*g* for 30 s and then precipitated in 10% (vol/vol) trichloroacetic acid at –20°C for 30 min. Precipitated

proteins were collected by centrifugation at 15,000×$g$ for 30 min at 4°C, and then washed twice with acetone. The air-dried pellets were resuspended in SDS-loading dye and used as the secreted protein samples. All the samples were boiled for 10 min before SDS-PAGE analysis.

## Western blotting analysis

Proteins were separated using SDS-PAGE and subsequently transferred onto a PVDF membrane (Bio-Rad). Prior to antibody incubation, the membrane was blocked with a solution of 5% (wt/vol) non-fat milk dissolved in TBST buffer (consisting of 50 mM Tris, 150 mM NaCl, 0.1% [vol/vol] Tween 20, pH 7.6) for 1 hr at room temperature. The membrane was then incubated sequentially with primary and secondary antibodies. The Clarity ECL solution (Bio-Rad) was utilized for signal detection. Monoclonal antibodies specific to RpoB (RNA polymerase beta subunit) were sourced from BioLegend (Product # 663905). Additionally, polyclonal antibodies to Hcp were customized by Shanghai Youlong Biotech (*Pei et al., 2020*). HRP-linked secondary antibodies were purchased from Beyotime Biotechnology (Product # A0208 and # A0216, respectively).

## Protein expression and purification

Proteins were expressed using pET28a vector with the His tag in *E. coli* BL21 (DE3). Strains containing the recombinant plasmid were grown in LB at 37°C. When the optical density of the culture reached OD$_{600}$ of 0.6, isopropyl-β-$_D$-thiogalactopyranoside was added to a final concentration of 1 mM, and the culture was incubated at 20°C for 16 hr. The cells were collected by centrifugation at 2500×$g$ for 10 min at 4°C, resuspended in lysis buffer (20 mM Tris-HCl, 300 mM NaCl, 10 mM imidazole, pH 8.0), and lysed by sonication. After sonication, the lysate was centrifuged at 12,000×$g$ for 60 min at 4°C, and the supernatant was collected. Ni-NTA affinity chromatography was used to purify the target protein, followed by gel filtration using Superdex 200 pg 10/300 column (GE Healthcare). The protein was finally concentrated to 15 mg/ml. All samples were visualized by SDS-PAGE and stained with Coomassie brilliant blue dye. Gel images for all purified proteins used in this study are shown in *Figure 6—figure supplement 3*.

## Crystallization, X-ray data collection, and structure determination

TseP$^C$ was crystallized at 18°C using the hanging drop method. Equal volumes (1:1) of the protein solution and reservoir solution, containing 0.1 M sodium formate (pH 7.0) and 12% (wt/vol) PEG 3350, were mixed. The crystals were cryoprotected in a mother liquor containing 20% vol/vol glycol and were flash-cooled at 100 K.

An X-ray diffraction dataset of TseP$^C$ was collected at beamline BL19U1 (wavelength, 0.97852 Å) at the Shanghai Synchrotron Radiation Facility. Diffraction data was auto-processed by an aquarium pipeline. The crystals were diffracted to 2.27 Å and belonged to the space group C 2 2 21 with unit cell dimensions of a=44.48 Å, b=138.8 Å, and c=95.83 Å. This structure was solved by molecular replacement with PHASER using the previously reported SPN1S endolysin structure (PDB ID: 4OK7) as the searching model and further polished by COOT and PHENIX. refine (*Adams et al., 2002*; *Emsley and Cowtan, 2004*). All structure figures were generated using PyMOL (https://pymol.org/2/) software (*Schrödinger, 2015*).

## PG purification

PG was extracted as previously described (*Glauner, 1988*). Bacteria were cultured at 37°C for 8 hr. Cells were then centrifuged at 4°C for 30 min at 3000×$g$ and the cell pellets were washed with ultra-pure water. After washing, the bacteria were resuspended in ultrapure water to an OD$_{600}$ of 70–100 and then added dropwise to an equal volume of boiling 8% SDS solution. The mixture was slowly stirred while boiling for 3 hr and then cooled naturally to room temperature. The precipitate was collected at 20°C for 1 hr at 100,000×$g$ and washed with ultrapure water five times. To remove the residual proteins and DNA, the precipitate was resuspended in 10 ml of 10 mM Tris-HCl (pH 7.0) and treated with 5 mg DNase I and 16 mg trypsin at 37°C for 16 hr. After treatment, the mixture was centrifuged at room temperature for 1 hr at 100,000×$g$, and the pellets were dissolved and treated with an equal volume of boiling 8% SDS solution for another 3 hr. The purified PG was obtained after washing with ultrapure water, freeze-dried, and stored at –20°C.

## In vitro enzymatic assays and enzyme kinetics

To characterize the in vitro enzymatic activity of TseP and its homologs, 500 μg of purified *E. coli* PG was co-incubated with an equal amount of proteins in a 0.1 ml reaction system at 37°C for 12 hr. After centrifugation, the supernatant was collected and subjected to analysis by ultraperformance liquid chromatography-quadrupole time of flight mass spectrometry (UPLC-QTOF-MS). The Acquity UPLC HSS T3 column was chosen as the solid phase, with solvent A (water with 0.1% [vol/vol] formic acid) and solvent B (methanol) as the mobile phase. Analyte separation was achieved by a linear gradient of solvent B, which was gradually increased from 1% to 20% over 65 min. Thereafter, a rapid gradient adjustment was followed to increase the concentration of solvent B from 20% to 100% in 1 min, with a retention period of 5 min to ensure complete separation. Finally, the system was flushed with 99% solvent A for 4 min to complete the process.

To measure the enzyme kinetics of TseP[C], a dye-release method was used to detect the PG hydrolyzing. Prior to use, the purified PG was labeled by Remazol Brilliant Blue (RBB) following a previously described method (*Yang et al., 2018*). Briefly, purified PG was incubated with 20 mM RBB in 0.25 M NaOH for 12 hr at 37°C. The RBB-labeled PG was collected via centrifugation for 10 min at 21,400×*g* and washed with ultrapure water several times until no RBB dye could be detected in the supernatant. This RBB-labeled PG was then applied for the enzyme kinetics analysis. In each reaction, 10 nM enzyme was mixed with different concentrations of RBB-labeled PG at 37°C for 10 min. Undigested PG was pelleted by centrifugation at 21,400×*g* for 10 min and the dye release in the supernatant was quantified by measuring its absorbance at $OD_{595}$.

## Bacterial competition assays

Killer and prey strains were grown aerobically in liquid LB medium until they reached $OD_{600}$ values of 1 and 2, respectively. Later, the cells were harvested via centrifugation and gently resuspended in fresh LB broth. Killer and prey cells mixed at a ratio of 5:1 were deposited onto LB-agar plates supplemented with 0.01% (wt/vol) L-arabinose and co-incubated for 6 hr at 28°C. Afterward, the mixed cells were recovered in fresh LB broth and vigorously vortexed to dislodge the agar. For the Gram-positive bacterial killing, the SSU and *B. subtilis* cells were mixed at a ratio of 20:1 with 0.01% L-arabinose in liquid LB medium for 24 hr at 28°C. The mixtures were then serially diluted and plated onto specific antibiotic-containing plates.

## Fluorescence microscopic imaging analysis

Bacteria grown to an $OD_{600}$ of 1.0 at 37°C were resuspended in fresh LB medium to achieve a final $OD_{600}$ of 10.0. A 1 μl aliquot was then spotted onto an agarose gel layer containing 0.01% L-arabinose, placed on a clean, dust-free slide, and observed under a microscope (Nikon, Ti2-E). The microscope settings were configured with a ×100 oil objective lens, and GFP signals were visualized using ET-GFP (Chroma 49002) parameters. T6SS assembly in bacteria was counted through sequential time-lapse photography over a period of 5 min, with an interval of 10 s between frames. At least three imaging fields were selected, and the number of cells and T6SS assemblies within a representative 30×30 μm² area in each field were counted.

## Bioinformatics analysis

The gene sequences of *A. dhakensis* SSU were retrieved from the draft genome assembly (GenBank accession NZ_JH815591.1) and confirmed by Sanger sequencing. The protein sequence of TseP was analyzed using blastp to identify homologs (*Altschul et al., 1990*). A representative set of the top 14 hits was downloaded from the UniProt (*Trifinopoulos et al., 2023*). Phylogeny was constructed using the IQ-tree web server with bootstrap 1000 times (*Trifinopoulos et al., 2016*). The resulting sequences were then aligned using Clustal Omega (*Sievers et al., 2011*), and the alignment was visualized using ESPript with its default settings (https://espript.ibcp.fr) (*Robert and Gouet, 2014*).

## Acknowledgements

This work was supported by funding from National Natural Science Foundation of China (32030001) and National Key R&D Program of China (2020YFA0907200). The funders had no role in study design, data collection and interpretation, or the decision to submit the work for publication.

## Additional information

### Funding

| Funder | Grant reference number | Author |
|---|---|---|
| National Natural Science Foundation of China | 32030001 | Tao Dong |
| National Key Research and Development Program of China | 2020YFA0907200 | Tao Dong |

The funders had no role in study design, data collection and interpretation, or the decision to submit the work for publication.

### Author contributions

Zeng-Hang Wang, Data curation, Formal analysis, Investigation, Methodology, Writing – original draft; Ying An, Tong-Tong Pei, Data curation, Investigation, Methodology; Ting Zhao, Data curation, Investigation; Dora Yuping Wang, Visualization; Xiaoye Liang, Investigation, Writing - review and editing; Wenming Qin, Supervision, Investigation; Tao Dong, Conceptualization, Supervision, Funding acquisition, Investigation, Methodology, Writing – original draft, Writing - review and editing

### Author ORCIDs

Zeng-Hang Wang ⬤ https://orcid.org/0009-0004-0532-1070
Ying An ⬤ http://orcid.org/0000-0002-8536-6954
Tong-Tong Pei ⬤ http://orcid.org/0009-0004-8241-8896
Tao Dong ⬤ https://orcid.org/0000-0003-3557-1850

Reviewer #1 (Public review): https://doi.org/10.7554/eLife.101125.3.sa1
Reviewer #2 (Public review): https://doi.org/10.7554/eLife.101125.3.sa2
Reviewer #3 (Public review): https://doi.org/10.7554/eLife.101125.3.sa3
Author response https://doi.org/10.7554/eLife.101125.3.sa4

## Additional files

### Supplementary files

Supplementary file 1. Strains and plasmids used in this study, and the sequence identity of the TseP and homologs. (A) Strains and plasmids used in this study. (B) The sequence identity of the N-terminal of TseP and homologs. Sequence identity was calculated using ClustalW web server. Colors are assigned from blue to red, indicating low to high sequence similarity, respectively. (C) The sequence identity of the C-terminal of TseP and homologs. Sequence identity was calculated using ClustalW web server. Colors are assigned from blue to red, indicating low to high sequence similarity, respectively.

Supplementary file 2. Homolog proteins with similar structures to TseP$^N$ in the Foldseek Search AFDB50 database.

Supplementary file 3. Homolog proteins with similar structures to TseP$^C$ in the Foldseek Search AFDB50 database.

MDAR checklist

### Data availability

The atomic coordinate and structure factor of TsePC have been deposited in the Protein Data Bank with accession code 8XCL.

The following dataset was generated:

| Author(s) | Year | Dataset title | Dataset URL | Database and Identifier |
|---|---|---|---|---|
| Wang ZH, Zhao T, Qin WM, Dong T | 2023 | Crystal structure of the C-terminal domain of the T6SS effector TseP from Aeromonas dhakensis SSU | https://www.rcsb.org/structure/8XCL | RCSB Protein Data Bank, 8XCL |

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
