## [Editor Report · eLife Assessment]

This **important** study describes how a single effector of the Type Six Secretion System (T6SS) has two distinct functions, which may contribute to bacterial survival and the development of novel antibacterials. The authors utilized various methods in biochemistry, microbiology, and microscopy to produce **convincing** data supporting their claims about the protein's function; however, they could clarify the implications for non-experts to enhance the accessibility of this work. This manuscript is of interest to those studying T6SS, particularly those interested in effectors and bacterial enzymes.

---

## [Referee Report · Reviewer #1 (Public review)]

Summary:

The manuscript performs a comprehensive biochemical, structural, and bioinformatic analysis of TseP, a type 6 secretion system effector from Aeromonas dhakensis that includes identification of a domain required for secretion and residues conferring target organism specificity. Through targeted mutations, they have expanded the target range of a T6SS effector to include a gram-positive species, which are not typically susceptible to T6SS attack. Although this is not the first dual domain effector to be described, this is the first time anyone has been able to modify a T6SS effector to have an expanded target species range.

Strengths:

The thorough dissection of TseP activity and modulation of target specificity represent a novel contribution to the field of antibacterial research.

Weaknesses:

Although the mechanistic activity of TseP is fully dissected here, there are some unaddressed questions regarding the importance/evolution of the dual activity domain organization. For example, does the modified Gram-positive targeting TseP effector still kill Gram-negative bacteria in bacterial mixtures? And if so, what is the evolutionary benefit of having a TseP that cannot target Gram-positives? And can something be inferred about the biology of Aeromonas from this?

Comments on revisions:

The comments and critiques from the initial submission have been addressed. However, some of them have only been addressed in the author's rebuttal. Some of the discussion particularly regarding the validity of using E. coli PG, the ability for TseP_C4+ to still kill *E. coli*, and the advantages of having dual domain function effectors probably should be present in the actual manuscript.

---

## [Referee Report · Reviewer #2 (Public review)]

Summary:

Wang et al. investigate the role of TseP, a Type VI secretion system (T6SS) effector molecule, revealing its dual enzymatic activities as both an amidase and a lysozyme. This discovery significantly enhances the understanding of T6SS effectors, which are known for their roles in interbacterial competition and survival in polymicrobial environments. TseP's dual function is proposed to play a crucial role in bacterial survival strategies, particularly in hostile environments where competition between bacterial species is prevalent.

Strengths:

(1) The dual enzymatic function of TseP is a significant contribution, expanding the understanding of T6SS effectors.

(2) The study provides important insights into bacterial survival strategies, particularly in interbacterial competition.

(3) The findings have implications for antimicrobial research and understanding bacterial interactions in complex environments.

Weaknesses:

(1) The manuscript assumes familiarity with previous work, making it difficult to follow. Mutants and strains need clearer definition and references.

(2) Figures lack proper controls, quantification, and clarity in some areas, notably in Figures 1A and 1C.

(3) The Materials and Methods section is poorly organized, hindering reproducibility. Biophysical validation of Zn²⁺ interaction and structural integrity of proteins need to be addressed.

(4) Discrepancies in protein degradation patterns and activities across different figures raise concerns about data reliability.

Comments on revisions:

The authors have addressed most of the comments, significantly improving the manuscript. They provided clear details of mutant constructs and strains, including additional references and a revised strain. Individual data points and statistical analyses were added to key figures, ensuring transparency and reproducibility. Supplemental data, such as protein purification details and loading controls, were included to address concerns about experimental reliability. However, the authors did not perform new experiments, such as isothermal titration calorimetry (ITC) to demonstrate the interaction between Zn^2+^ and TsePN or stop-flow spectroscopy to examine enzymatic kinetics, which could have further strengthened the manuscript. I trust these aspects will be addressed in future studies.

The revised Materials and Methods section was significantly improved, providing detailed protocols for bioinformatics analyses, microscopic imaging, and enzymatic assays.

These revisions provide a clearer and more robust presentation of TseP's dual enzymatic functions and their implications in bacterial competition. The manuscript now represents a significant contribution to understanding T6SS effectors, and I recommend it for publication in its current form.

---

## [Referee Report · Reviewer #3 (Public review)]

Summary:

Type VI secretion systems (T6SS) are employed by bacteria to inject competitor cells with numerous effector proteins. These effectors can kill injected cells via an array of enzymatic activities. A common class of T6SS effector are peptidoglycan (PG) lysing enzymes. In this manuscript, the authors characterize a PG-lysing effector-TseP-from the pathogen Aeromonas dhakensis. While the C-terminal domain of TseP was known to have lysozyme activity, the N-terminal domain was uncharacterized. Here, the authors functionally characterize TsePN as a zinc-dependent amidase. This discovery is somewhat novel because it is rare for PG-lysing effectors to have amidase and lysozyme activity. In the second half of the manuscript, the authors utilize a crystal structure of the lysozyme TsePC domain to inform the engineering of this domain to lyse gram-positive peptidoglycan.

Strengths:

The two halves of the manuscript considered together provide a nice characterization of a unique T6SS effector and reveal potentially general principles for lysozyme engineering.

Weaknesses:

The advantage of fusing amidase and lysozyme domains in a single effector is not discussed but would appear to be a pertinent question.

Comments on revisions:

The authors have adequately addressed my previous comments. The authors did not conduct any additional experiments to address the comments made by other reviewers. However, in most cases it seems that paring down the strength of claims made in the text or adding data to the supplement is sufficient to address these concerns.

---

## [Author Response]

The following is the authors’ response to the original reviews.

**Public Reviews:**

**Reviewer #1 (Public review):**
Summary:The manuscript performs a comprehensive biochemical, structural, and bioinformatic analysis of TseP, a type 6 secretion system effector from Aeromonas dhakensis that includes the identification of a domain required for secretion and residues conferring target organism specificity. Through targeted mutations, they have expanded the target range of a T6SS effector to include a gram-positive species, which is not typically susceptible to T6SS attack.Strengths:All of the experiments presented in the study are well-motivated and the conclusions are generally sound.

Thank you.

Weaknesses:There are some issues with the clarity of figures. For example, the microscopy figures could have been more clearly presented as cell counts/quantification rather than representative images. Similarly, loading controls for the secreted proteins for the westerns probably should be shown.Also, some of the minor/secondary conclusions reached regarding the "independence" of the N and C term domains of the TseP are a bit overreaching.

We thank the reviewer for pointing out the issues and have carefully revised the manuscript accordingly. We acknowledge the reviewer’s concern regarding the independence of the N- and C-terminal domains, and have toned down the relevant claims.

**Reviewer #2 (Public review):**
Summary:Wang et al. investigate the role of TseP, a Type VI secretion system (T6SS) effector molecule, revealing its dual enzymatic activities as both an amidase and a lysozyme. This discovery significantly enhances the understanding of T6SS effectors, which are known for their roles in interbacterial competition and survival in polymicrobial environments. TseP's dual function is proposed to play a crucial role in bacterial survival strategies, particularly in hostile environments where competition between bacterial species is prevalent.Strengths:(1) The dual enzymatic function of TseP is a significant contribution, expanding the understanding of T6SS effectors.(2) The study provides important insights into bacterial survival strategies, particularly in interbacterial competition.(3) The findings have implications for antimicrobial research and understanding bacterial interactions in complex environments.

Thank you.

Weaknesses:(1) The manuscript assumes familiarity with previous work, making it difficult to follow. Mutants and strains need clearer definitions and references.

Thank you for raising the issue. We have revised the manuscript accordingly to improve the clarity by including more detailed descriptions of the mutants and strains, along with references to prior work where relevant, to improve clarity.

(2) Figures lack proper controls, quantification, and clarity in some areas, notably in Figures 1A and 1C.

We have now added the controls as requested by reviewers.

(3) The Materials and Methods section is poorly organized, hindering reproducibility. Biophysical validation of Zn^2+^ interaction and structural integrity of proteins need to be addressed.

We have now included more details in the Materials and Methods section. While we recognize the importance of biophysical validation of the Zn^2+^ interaction, this analysis lies beyond the primary scope of the current study. We plan to investigate the role of Zn²⁺ interaction and the EF-hand domain in greater depth as part of our follow-up studies. Thank you for this suggestion.

(4) Discrepancies in protein degradation patterns and activities across different figures raise concerns about data reliability.

We acknowledge the concern about discrepancies in protein degradation patterns. TseP exhibits inherent instability, which might explain the observed variations. We have added an explanation in the detailed response letter and the manuscript.

**Reviewer #3 (Public review):**
Summary:Type VI secretion systems (T6SS) are employed by bacteria to inject competitor cells with numerous effector proteins. These effectors can kill injected cells via an array of enzymatic activities. A common class of T6SS effector are peptidoglycan (PG) lysing enzymes. In this manuscript, the authors characterize a PG-lysing effector-TseP-from the pathogen Aeromonas dhakensis. While the C-terminal domain of TseP was known to have lysozyme activity, the N-terminal domain was uncharacterized. Here, the authors functionally characterize TsePN as a zinc-dependent amidase. This discovery is somewhat novel because it is rare for PG-lysing effectors to have amidase and lysozyme activity.In the second half of the manuscript, the authors utilize a crystal structure of the lysozyme TsePC domain to inform the engineering of this domain to lyse gram-positive peptidoglycan.Strengths:The two halves of the manuscript considered together provide a nice characterization of a unique T6SS effector and reveal potentially general principles for lysozyme engineering.

Thank you.

Weaknesses:The advantage of fusing amidase and lysozyme domains in a single effector is not discussed but would appear to be a pertinent question. Labeling of the figures could be improved to help readers understand the data.

Thank you for the suggestions. We have revised the manuscript and figures to improve clarity.

The advantage of having dual-domain functions relative to having just one of the two functions is likely for increasing competitive fitness. Although such dual functional cell-wall targeting effectors have not been characterized prior to this study, there are some examples that dual functions are encoded by the same secretion module, for example the VgrG1-TseL pair in *Vibrio cholerae*. The C-terminal of VgrG1 not only catalyzes actin crosslinking but also recognizes and delivers the downstream encoded lipase effector TseL through direct interaction. In this context, the VgrG1-TseL pair also represent a dual-functional module. Therefore, it is likely that fusing effector domains and coupling effector functions are parallel strategies for the evolution of T6SS effectors.